# Global Sensitivity Analysis
# for MAP Inference in Graphical Models

**Jasper De Bock**
Ghent University, SYSTeMS
Ghent (Belgium)
jasper.debock@ugent.be

**Cassio P. de Campos**
Queen's University
Belfast (UK)
c.decampos@qub.ac.uk

**Alessandro Antonucci**
IDSIA
Lugano (Switzerland)
alessandro@idsia.ch

## Abstract

We study the sensitivity of a MAP configuration of a discrete probabilistic graphical model with respect to perturbations of its parameters. These perturbations are global, in the sense that simultaneous perturbations of all the parameters (or any chosen subset of them) are allowed. Our main contribution is an exact algorithm that can check whether the MAP configuration is robust with respect to given perturbations. Its complexity is essentially the same as that of obtaining the MAP configuration itself, so it can be promptly used with minimal effort. We use our algorithm to identify the largest global perturbation that does not induce a change in the MAP configuration, and we successfully apply this robustness measure in two practical scenarios: the prediction of facial action units with posed images and the classification of multiple real public data sets. A strong correlation between the proposed robustness measure and accuracy is verified in both scenarios.

## 1 Introduction

*Probabilistic graphical models* (PGMs) such as *Markov random fields* (MRFs) and *Bayesian networks* (BNs) are widely used as a knowledge representation tool for reasoning under uncertainty. When coping with such a PGM, it is not always practical to obtain numerical estimates of the parameters—the local probabilities of a BN or the factors of an MRF—with sufficient precision. This is true even for quantifications based on data, but it becomes especially important when eliciting the parameters from experts. An important question is therefore how precise these estimates should be to avoid a degradation in the diagnostic performance of the model. This remains important even if the accuracy can be arbitrarily refined in order to trade it off with the relative costs. This paper is an attempt to systematically answer this question.

More specifically, we address *sensitivity analysis* (SA) of discrete PGMs in the case of *maximum a posteriori* (MAP) inferences, by which we mean the computation of the most probable configuration of some variables given an observation of all others.[1]

Let us clarify the way we intend SA here, while giving a short overview of previous work on SA in PGMs. First of all, a distinction should be made between *quantitative* and *qualitative* SA. Quantitative approaches are supposed to evaluate the effect of a perturbation of the parameters on the numerical value of a particular inference. Qualitative SA is concerned with deciding whether or not the perturbed values are leading to a different decision, e.g., about the most probable configuration of the queried variable(s). Most of the previous work in SA is quantitative, being in particular focused on *updating*, i.e., the computation of the posterior probability of a single variable given some evidence, and mostly focus on BNs. After a first attempt based on a purely empirical investigation [17], a number of analytical methods based on the derivatives of the updated probability with respect to

the perturbed parameters have been proposed [3, 4, 5, 11, 14]. Something similar has been done for MRFs as well [6]. To the best of our knowledge, qualitative SA received almost no attention, with few exceptions [7, 18].

Secondly, we distinguish between *local* and *global* SA. The former considers the effect of the perturbation of a single parameter (and of possible additional perturbations that are induced by normalization constraints), while the latter aims at more general perturbations possibly affecting all the parameters of the PGM. Initial work on SA in PGMs considered the local approach [4, 14], while later work considered global SA as well [3, 5, 11]. Yet, for BNs, global SA has been tackled by methods whose time complexity is exponential in the number of perturbed *conditional probability tables* (CPTs), as they basically require the computation of all the mixed derivatives. For qualitative SA, as far as we know, only the local approach has been studied [7, 18]. This is unfortunate, as global SA might reveal stronger effects of perturbations due to synergetic effects, which might remain hidden in a local analysis.

In this paper, we study global qualitative SA in discrete PGMs for MAP inferences, thereby intending to fill the existing gap in this topic. Let us introduce it by a simple example.

**Example 1.** *Let $X_1$ and $X_2$ be two Boolean variables. For each $i \in \{1, 2\}$, $X_i$ takes values in $\{x_i, \neg x_i\}$. The following probabilistic assessments are available: $P(x_1) = .45$, $P(x_2|x_1) = .2$, and $P(x_2|\neg x_1) = .9$. This induces a complete specification of the joint probability mass function $P(X_1, X_2)$. If no evidence is present, the MAP joint state is $(\neg x_1, x_2)$, its probability being .495. The second most probable joint state is $(x_1, \neg x_2)$, whose probability is .36. We perturb the above three parameters. Given $\epsilon_{x_1} \geq 0$, we consider any assessment of $P(x_1)$ such that $|P(x_1) - .45| \leq \epsilon_{x_1}$. We similarly perturb $P(x_2|x_1)$ with $\epsilon_{x_2|x_1}$ and $P(x_2|\neg x_1)$ with $\epsilon_{x_2|\neg x_1}$. The goal is to investigate whether or not $(\neg x_1, x_2)$ is also the unique MAP instantiation for each $P(X_1, X_2)$ consistent with the above constraints, given a maximum perturbation level of $\epsilon = .06$ for each parameter. Straightforward calculations show that this is true if only one parameter is perturbed at each time. The state $(\neg x_1, x_2)$ remains the most probable even if two parameters are perturbed (for any pair of them). The situation is different if the perturbation level $\epsilon = .06$ is applied to all three parameters simultaneously. There is a specification of the parameters consistent with the perturbations and such that the MAP instantiation is $(x_1, \neg x_2)$ and achieves probability .4386, corresponding to $P(x_1) = .51$, $P(x_2|x_1) = .14$, and $P(x_2|\neg x_1) = .84$. The minimum perturbation level for which this behaviour is observed is $\epsilon^* = .05$. For this value, there is a single specification of the model for which $(x_1, \neg x_2)$ has the same probability as $(\neg x_1, x_2)$, which—for this value—is the single most probable instantiation for any other specification of the model that is consistent with the perturbations.*

The above example can be regarded as a qualitative SA for which the local approach is unable to identify a lack of robustness in the MAP solution, which is revealed instead by the global analysis. In the rest of the paper we develop an algorithm to efficiently detect the minimum perturbation level $\epsilon^*$ leading to a different MAP solution. The time complexity of the algorithm is equal to that of the MAP inference in the PGM times the number of variables in the domain, that is, exponential in the treewidth of the graph in the worst case. The approach can be specialized to local SA or any other choice of parameters to perform SA, thus reproducing and extending existing results. The paper is organized as follows: the problem of checking the robustness of a MAP inference is introduced in its general formulation in Section 2. The discussion is then specialized to the case of PGMs in Section 3 and applied to global SA in Section 4. Experiments with real data sets are reported in Section 5, while conclusions and outlooks are given in Section 6.

## 2 MAP Inference and its Robustness

We start by explaining how we intend SA for MAP inference and how this problem can be translated into an optimisation problem very similar to that used for the computation of MAP itself. For the sake of readibility, but without any lack of generality, we begin by considering a single variable only; the multivariate and the conditional cases are dicussed in Section 3. Consider a single variable $X$ taking its values in a finite set $\text{Val}(X)$. Given a probability mass function $P$ over $X$, $\tilde{x} \in \text{Val}(X)$ is said to be a *MAP instantiation* for $P$ if

$$\tilde{x} \in \arg \max_{x \in \text{Val}(X)} P(x), \tag{1}$$

which means that $\tilde{x}$ is the most likely value of $X$ according to $P$. In principle a mass function $P$ can have multiple (equally probable) MAP instantiations. However, in practice there will often be only one, and we then call it the *unique MAP instantiation* for $P$.

As we did in Example 1, SA can be achieved by modeling perturbations of the parameters in terms of (linear) constraints over them, which are used to define the set of all perturbed models whose mass function is consistent with these constraints. Generally speaking, we consider an arbitrary set $\mathcal{P}$ of *candidate* mass functions, one of which is the original unperturbed mass function $P$. The only imposed restriction is that $\mathcal{P}$ must be compact. This way of defining candidate models establishes a link between SA and the theory of *imprecise probability*, which extends the Bayesian theory of probability to cope with compact (and often convex) sets of mass functions [19].

For the MAP inference in Eq. (1), performing SA with respect to a set of candidate models $\mathcal{P}$ requires the identification of the instantiations that are MAP for at least one perturbed mass function, that is,

$$\mathrm{Val}^*(X) := \left\{ \tilde{x} \in \mathrm{Val}(X) \ \middle| \ \exists P' \in \mathcal{P} : \tilde{x} \in \arg \max_{x \in \mathrm{Val}(X)} P'(x) \right\}. \tag{2}$$

These instantiations are called *E-admissible* [15]. If the above set contains only a single MAP instantiation $\tilde{x}$ (which is then necessarily the unique solution of Eq. (1) as well), then we say that the model $P$ is *robust* with respect to the perturbation $\mathcal{P}$.

**Example 2.** *Let $X$ take values in $\mathrm{Val}(X) := \{a, b, c, d\}$. Consider a perturbation $\mathcal{P} := \{P_1, P_2\}$ that contains only two candidate mass functions over $X$. Let $P_1$ be defined by $P_1(a) = .5$, $P_1(b) = P_1(c) = .2$ and $P_1(d) = .1$ and let $P_2$ be defined by $P_2(b) = .35$, $P_2(a) = P_2(c) = .3$ and $P_2(d) = .05$. Then $a$ and $b$ are the unique MAP instantiations of $P_1$ and $P_2$, respectively. This implies that $\mathrm{Val}^*(X) = \{a, b\}$ and that neither $P_1$ nor $P_2$ is robust with respect to $\mathcal{P}$.*

For large domains $\mathrm{Val}(X)$, for instance in the multivariate case, evaluating $\mathrm{Val}^*(X)$ is a time-consuming task that is often intractable. However, if we are not interested in evaluating $\mathrm{Val}^*(X)$, but only want to decide whether or not $P$ is robust with respect to the perturbation described by $\mathcal{P}$, more efficient methods can be used. The following theorem establishes how this decision can be reformulated as an optimisation problem that, as we are about to show in Section 3, can be solved efficiently for PGMs. Due to space constraints, the proofs are provided as supplementary material.

**Theorem 1.** *Let $X$ be a variable taking values in a finite set $\mathrm{Val}(X)$ and let $\mathcal{P}$ be a set of candidate mass functions over $X$. Let $\tilde{x}$ be a MAP instantiation for a mass funtion $P \in \mathcal{P}$. Then $\tilde{x}$ is the unique MAP instantiation for every $P' \in \mathcal{P}$, that is, $\mathrm{Val}^*(X)$ has cardinality one, if and only if*

$$\min_{P' \in \mathcal{P}} P'(\tilde{x}) > 0 \ \ \text{and} \ \ \max_{x \in \mathrm{Val}(X) \setminus \{\tilde{x}\}} \max_{P' \in \mathcal{P}} \frac{P'(x)}{P'(\tilde{x})} < 1, \tag{3}$$

*where the first inequality should be checked first because if it fails, then the left-hand side of the second inequality is ill-defined.*

## 3 PGMs and Efficient Robustness Verification

Let $\mathbf{X} = (X_1, \ldots, X_n)$ be a vector of variables taking values in their respective finite domains $\mathrm{Val}(X_1), \ldots, \mathrm{Val}(X_n)$. We will use $[n]$ a shorthand notation for $\{1, \ldots, n\}$, and similarly for other natural numbers. For every non-empty $C \subseteq [n]$, $\mathbf{X}_C$ is a vector that consists of the variables $X_i$, $i \in C$, that takes values in $\mathrm{Val}(\mathbf{X}_C) := \times_{i \in C} \mathrm{Val}(X_i)$. For $C = [n]$ and $C = \{i\}$, we obtain $\mathbf{X} = \mathbf{X}_{[n]}$ and $X_i = \mathbf{X}_{\{i\}}$ as important special cases. A *factor* $\phi$ over a vector $\mathbf{X}_C$ is a real-valued map on $\mathrm{Val}(\mathbf{X}_C)$. If for all $\mathbf{x}_C \in \mathbf{X}_C$, $\phi(\mathbf{x}_C) \geq 0$, then $\phi$ is said to be *nonnegative*.

Let $I_1, \ldots, I_m$ be a collection of index sets such that $I_1 \cup \cdots \cup I_m = [n]$ and $\Phi = \{\phi_1, \ldots, \phi_m\}$ be a set of nonnegative factors over the vectors $\mathbf{X}_{I_1}, \ldots, \mathbf{X}_{I_m}$, respectively. We say that $\Phi$ is a PGM if it induces a joint probability mass function $P_\Phi$ over $\mathrm{Val}(\mathbf{X})$, defined by

$$P_\Phi(\mathbf{x}) := \frac{1}{Z_\Phi} \prod_{k=1}^m \phi_k(\mathbf{x}_{I_k}) \text{ for all } \mathbf{x} \in \mathrm{Val}(\mathbf{X}), \tag{4}$$

where $Z_\Phi := \sum_{\mathbf{x} \in \mathrm{Val}(\mathbf{X})} \prod_{k=1}^m \phi_k(\mathbf{x}_{I_k})$ is the normalising constant called partition function. Since $\mathrm{Val}(\mathbf{X})$ is finite, $\Phi$ is a PGM if and only if $Z_\Phi > 0$.

## 3.1 MAP and Second Best MAP Inference for PGMs

If $\Phi$ is a PGM then, by merging Eqs. (1) and (4), we see that $\tilde{\mathbf{x}} \in \mathrm{Val}(\mathbf{X})$ is a MAP instantiation for $P_\Phi$ if and only if

$$\prod_{k=1}^{m} \phi_k(\mathbf{x}_{I_k}) \leq \prod_{k=1}^{m} \phi_k(\tilde{\mathbf{x}}_{I_k}) \text{ for all } \mathbf{x} \in \mathrm{Val}(\mathbf{X}),$$

where $\tilde{\mathbf{x}}_{I_k}$ is the unique element of $\mathrm{Val}(\mathbf{X}_{I_k})$ that is consistent with $\tilde{\mathbf{x}}$, and likewise for $\mathbf{x}_{I_k}$ and $\mathbf{x}$.

Similarly, $\mathbf{x}^{(2)} \in \mathrm{Val}(\mathbf{X})$ is said to be a *second best MAP instantiation* for $P_\Phi$ if and only if there is a MAP instantiation $\mathbf{x}^{(1)}$ for $P_\Phi$ such that $\mathbf{x}^{(1)} \neq \mathbf{x}^{(2)}$ and

$$\prod_{k=1}^{m} \phi_k(\mathbf{x}_{I_k}) \leq \prod_{k=1}^{m} \phi_k(\mathbf{x}_{I_k}^{(2)}) \text{ for all } \mathbf{x} \in \mathrm{Val}(\mathbf{X}) \setminus \{\mathbf{x}^{(1)}\}. \tag{5}$$

MAP inference in PGMs is an NP-hard task (see [12] for details). The task can be solved exactly by junction tree algorithms in time exponential in the *treewidth* of the network's moral graph. While finding the k-th best instantiation might be an even harder task [13] for general $k$, the second best MAP instantiation can be found by a sequence of MAP queries: (i) compute a first best MAP instantiation $\tilde{\mathbf{x}}^{(1)}$; (ii) for each queried variable $X_i$, take the original PGM and add an extra factor for $X_i$ that equals 1 minus the indicator of the value that $X_i$ has in $\tilde{\mathbf{x}}^{(1)}$, and run the MAP inference; (iii) report the instantiation with highest probability among all these runs. Because the second best has to differ from the first best in at least one $X_i$ (and this is ensured by that extra factor), this procedure is correct and in worst case it spends time equal to a single MAP inference multiplied by the number of variables. Faster approaches to directly compute the second best MAP, without reduction to standard MAP queries, have been also proposed (see [8] for an overview).

## 3.2 Evaluating the Robustness of MAP Inference With Respect to a Family of PGMs

For every $k \in [m]$, let $\psi_k$ be a set of nonnegative factors over the vector $\mathbf{X}_{I_k}$. Every combination of factors $\Phi = \{\phi_1, \ldots, \phi_m\}$ from the sets $\psi_1, \ldots, \psi_m$, respectively, is called a *selection*. Let $\Psi := \times_{k=1}^{m} \psi_k$ be the set consisting of all these selections. If every selection $\Phi \in \Psi$ is a PGM, then $\Psi$ is said to be a *family* of PGMs. We then denote the corresponding set of distributions by $\mathcal{P}_\Psi := \{P_\Phi : \Phi \in \Psi\}$. In the following theorem, we establish that evaluating the robustness of MAP inference with respect to this set $\mathcal{P}_\Psi$ can be reduced to a second best MAP instantiation problem.

**Theorem 2.** *Let $\mathbf{X} = (X_1, \ldots, X_n)$ be a vector of variables taking values in their respective finite domains $\mathrm{Val}(X_1), \ldots, \mathrm{Val}(X_n)$, let $I_1, \ldots, I_m$ be a collection of index sets such that $I_1 \cup \cdots \cup I_m = [n]$ and, for every $k \in [m]$, let $\psi_k$ be a compact set of nonnegative factors over $\mathbf{X}_{I_k}$ such that $\Psi = \times_{k=1}^{m} \psi_k$ is a family of PGMs.*

*Consider now a PGM $\Phi \in \Psi$ and a MAP instantiation $\tilde{\mathbf{x}}$ for $P_\Phi$ and define, for every $k \in [m]$ and every $\mathbf{x}_{I_k} \in \mathrm{Val}(\mathbf{X}_{I_k})$:*

$$\alpha_k := \min_{\phi_k' \in \psi_k} \phi_k'(\tilde{\mathbf{x}}_k) \text{ and } \beta_k(\mathbf{x}_{I_k}) := \max_{\phi_k' \in \psi_k} \frac{\phi_k'(\mathbf{x}_{I_k})}{\phi_k'(\tilde{\mathbf{x}}_{I_k})}. \tag{6}$$

*Then $\tilde{\mathbf{x}}$ is the unique MAP instantiation for every $P' \in \mathcal{P}_\Psi$ if and only if*

$$(\forall k \in [m]) \; \alpha_k > 0 \text{ and } \prod_{k=1}^{m} \beta_k(\mathbf{x}_{I_k}^{(2)}) < 1, \tag{RMAP}$$

*where $\mathbf{x}^{(2)}$ is an arbitrary second best MAP instantiation for the distribution $P_{\tilde{\Phi}}$ that corresponds to the PGM $\tilde{\Phi} := \{\beta_1, \ldots, \beta_m\}$. The first criterion in (RMAP) should be checked first because $\beta_k(\mathbf{x}_{I_k}^{(2)})$ is ill-defined if $\alpha_k = 0$.*

Theorem 2 provides an algorithm to test the robustness of MAP in PGMs. From a computational point of view, checking (RMAP) can be done as described in the previous subsection, apart from the local computations appearing in Eq. (6). These local computations will depend on the particular choice of perturbation. As we will see further on, many natural perturbations induce very efficient local computations (usually because they are related somehow to simple linear or convex programming problems).

In most practical situations, some variables $\mathbf{X}_O$, with $O \subset [n]$, are observed and therefore known to be in a given configuration $\mathbf{y} \in \text{Val}(\mathbf{X}_O)$. In this case, the MAP inference for the conditional mass function $P_\Phi(\mathbf{X}_Q|\mathbf{y})$ should be considered, where $\mathbf{X}_Q := \mathbf{X}_{[n]\setminus O}$ are the queried variables. While we have avoided the discussion about the conditional case and considered only the MAP inference (and its robustness check) for the whole set of variables of the PGM, the standard technique employed with MRFs of including additional identity functions to encode observations suffices, as the probability of the observation (and therefore also the partition function value) does not influence the result of MAP inferences. Hence, one can run the MAP inference for the PGM $\Phi'$ augmented with local identity functions that yield $\mathbf{y}$, such that $Z_{\Phi'}P_{\Phi'}(\mathbf{X}_Q) = Z_\Phi P_\Phi(\mathbf{X}_Q, \mathbf{y})$ (that is, the unnormalized probabilities are equal, so MAP instantiations are equal too) and hence the very same techniques explained for the unconditional case are applicable to conditional MAP inference (and its robustness check) as well.

## 4 Global SA in PGMs

The most natural way to perform global SA in a PGM $\Phi = \{\phi_1, \ldots, \phi_m\}$ is by perturbing all its factors. Following the ideas introduced in Section 2 and 3, we model the effect of the perturbation by replacing the factor $\phi_k$ with a compact set $\psi_k$ of factors, for each $k \in [m]$. This induces a family $\Psi$ of PGMs. The condition (RMAP) can be therefore used to decide whether or not the MAP instantiation for $P_\Phi$ is the unique MAP instantiation for every $P' \in \mathcal{P}_\Psi$. In other words, we have an algorithm to test the robustness of $P_\Phi$ with respect to the perturbation $\mathcal{P}_\Psi$.

To characterize the perturbation level we introduce the notion of a *parametrized perturbation* $\psi_k^\epsilon$ of a factor $\phi_k$, defined by requiring that: (i) for each $\epsilon \in [0, 1]$, $\psi_k^\epsilon$ is a compact set of factors, each of which has the same domain as $\phi_k$; (ii) if $\epsilon_2 \geq \epsilon_1$, then $\psi_k^{\epsilon_2} \supseteq \psi_k^{\epsilon_1}$; and (iii) $\psi_k^0 = \{\phi_k\}$. Given a parametrized perturbation for each factor of the PGM $\Phi$, we denote by $\Psi^\epsilon$ the corresponding family of PGMs and by $\mathcal{P}_{\Psi^\epsilon}$ the relative set of joint mass functions.

We define the *critical perturbation threshold* $\epsilon^*$ as the supremum value of $\epsilon \in [0, 1]$ such that $P_{\Phi^\epsilon}$ is robust with respect to the perturbation $\mathcal{P}_{\Psi^\epsilon}$, i.e., such that the condition (RMAP) is still satisfied. Because of the property (ii) of parametrized perturbations, we know that if (RMAP) is not satisfied for a particular value of $\epsilon$ then it cannot be satisfied for a larger value and, vice versa, if the criterion is satisfied for a particular value than it will also be satisfied for every smaller value. An algorithm to evaluate $\epsilon^*$ can therefore be obtained by iteratively checking (RMAP) according to a bracketing scheme (e.g., bisection) over $\epsilon$. Local SA, as well as SA of only a selective collection of parameters, come as a byproduct, as one can perturb only some factors and our results and algorithm still apply.

### 4.1 Global SA in Markov Random Fields (MRFs)

MRFs are PGMs based on undirected graphs. The factors are associated to cliques of the graph. The specialization of the technique outlined by Theorem 2 is straightforward. A possible perturbation technique is the *rectangular* one. Given a factor $\phi_k$, its rectangular parametric perturbation $\psi_k^\epsilon$ is:

$$\psi_k^\epsilon = \{\phi_k' \geq 0 : |\phi_k'(\mathbf{x}_{I_k}) - \phi_k(\mathbf{x}_{I_k})| \leq \epsilon\Delta \text{ for all } \mathbf{x}_{I_k} \in \text{Val}(\mathbf{X}_{I_k})\}, \tag{7}$$

where $\Delta > 0$ is a chosen maximum perturbation level, achieved for $\epsilon = 1$.

For this kind of perturbation, the optimization in Eq. (6) is trivial: $\alpha_k = \max\{0, \phi_k(\tilde{\mathbf{x}}_k) - \epsilon\Delta\}$ and, if $\alpha_k > 0$, then $\beta_k(\tilde{\mathbf{x}}_{I_k}) = 1$ and, for all $\mathbf{x}_{I_k} \in \text{Val}(\mathbf{X}_{I_k}) \setminus \{\tilde{\mathbf{x}}_{I_k}\}$, $\beta_k(\mathbf{x}_{I_k}) = \frac{\phi_k(\mathbf{x}_{I_k}) + \epsilon\Delta}{\phi_k(\tilde{\mathbf{x}}_{I_k}) - \epsilon\Delta}$. If $\alpha_k = 0$, even for a single $k$, the criterion (RMAP) is not satisfied and $\beta_k$ should not be computed.

### 4.2 Global SA in Bayesian Networks (BNs)

BNs are PGMs based on directed graphs. The factors are CPTs, one for each variable, each conditioned on the parents of the variable. Each CPT contains a conditional mass function for each joint state of the parents. Perturbations in BNs can take this into consideration and use perturbations with a direct probabilistic interpretation. Consider an unconditional mass function $P$ over $X$. A parametrized perturbation $\mathcal{P}^\epsilon$ of $P$ can be achieved by $\epsilon$-contamination [2]:

$$\mathcal{P}^\epsilon := \{(1-\epsilon)P(X) + \epsilon P^*(X) : P^*(X) \text{ any mass function on } X\}. \tag{8}$$

It is a trivial exercise to check that this is a proper parametric perturbation of $P(X)$ and that $\mathcal{P}^1$ is the whole probabilistic simplex.

We perturb the CPTs of a BN by applying this parametric perturbation to every conditional mass function. Let $P(X|\mathbf{Y}) =: \psi(X, \mathbf{Y})$ be a CPT. The optimization in Eq. (6) is trivial also in this case. We have $\alpha_k = (1-\epsilon)P(\tilde{x}|\tilde{y})$ and, if $\alpha_k > 0$, then $\beta_k(\tilde{\mathbf{x}}_{I_k}) = 1$ and, for all $\mathbf{x}_{I_k} \in \mathrm{Val}(\mathbf{X}_{I_k}) \setminus \{\tilde{\mathbf{x}}_{I_k}\}$, $\beta_k(\mathbf{x}_{I_k}) = \frac{(1-\epsilon)P(x|\mathbf{y})+\epsilon}{(1-\epsilon)P(\tilde{x}|\tilde{\mathbf{y}})}$, where $\tilde{x}$ and $\tilde{\mathbf{y}}$ are consistent with $\tilde{\mathbf{x}}_{I_k}$ and similarly for $x$, $\mathbf{y}$ and $\mathbf{x}_{I_k}$.

More general perturbations can also be considered, and the efficiency of their computation relates to the optimization in Eq. (6). Because of that, we are sure that at least any linear or convex perturbation can be solved efficiently and in polynomial time by convex programming methods, while other more sophisticated perturbations might demand general non-linear optimization and hence cannot anymore ensure that computations are exact and quick.

## 5 Experiments

### 5.1 Facial Action Unit Recognition

We consider the problem of recognizing facial action units from real image data using the CK+ data set [10, 16]. Based on the Facial Action Coding System [9], facial behaviors can be decomposed into a set of 45 *action units* (AUs), which are related to contractions of specific sets of facial muscles. We work with 23 recurrent AUs (for a complete description, see [9]). Some AUs happen together to show a meaningful facial expression: AU6 (cheek raiser) tends to occur together with AU12 (lip corner puller) when someone is smiling. On the other hand, some AUs may be mutually exclusive: AU25 (lips part) never happens simultaneously with AU24 (lip presser) since they are activated by the same muscles but with opposite motions. The data set contains 68 landmark positions (given by coordinates $x$ and $y$) of the face of 589 posed individuals (after filtering out cases with missing data), as well as the labels for the AUs. Our goal is to predict all the AUs happening in a given image. In this work, we do not aim to outperform other methods designed for this particular task, but to analyse the robustness of a model when applied in this context. In spite of that, we expected to obtain a reasonably good accuracy by using an MRF.

One third of the posed faces are selected for testing, and two thirds for training the model. The labels of the testing data are not available during training and are used only to compute the accuracy of the predictions. Using the training data and following the ideas in [16], we build a linear *support vector machine* (SVM) separately for each one of the 23 AUs, using the image landmarks to predict that given AU. With these SVMs, we create new variables o1,. . ., o45, one for each selected AU, containing the predicted value from the SVM. This is performed for all the data, including training and testing data. After that, landmarks are discarded and the data is considered to have 46 variables (true values and SVM predicted ones). At this point, the accuracy of the SVM measurements on the testing data, if one considers the average Hamming distance between the vector of 23 true values and the vector of 23 predicted ones (that is, the sum of the number of times AU$i$ equals o$i$ over all $i$ and all instances in the testing data divided by 23 times the number of instances), is about 87%. We now use these 46 variables to build an MRF (we use a very simplistic penalized likelihood approach for learning the MRF, as the goal is not to obtain state-of-the-art classification but to analyse robustness), as shown in Fig. 1(a), where SVM-built variables are treated as observational/measurement nodes and relations are learned between the AUs (non displayed AU variables in the figure are only connected to their corresponding measurements).

Using the MRF, we predict the AU configuration using a MAP algorithm, where all AUs are queried and all measurement nodes are observed. As before, we characterise the accuracy of this model by the average Hamming distance between predicted vectors and true vectors, obtaining about 89% accuracy. That is, the inclusion of the relations between AUs by means of the MRF was able to slightly improve the accuracy obtained independently for each AU from the SVM. For our present purposes, we are however more interested in the associated perturbation thresholds $\epsilon^*$. For each instance of the testing data (that is, for each vector of 23 measurements), we compute it using the rectangular perturbations of Section 4.1. The higher $\epsilon^*$ is, the more robust is the issued vector, because it represents the single optimal MAP instantiation even if one varied all the parameters of the MRF by $\epsilon^*$. To understand the relation between $\epsilon^*$ and the accuracy of predictions, we have split the testing instances into bins, according to the Hamming distance between true and predicted

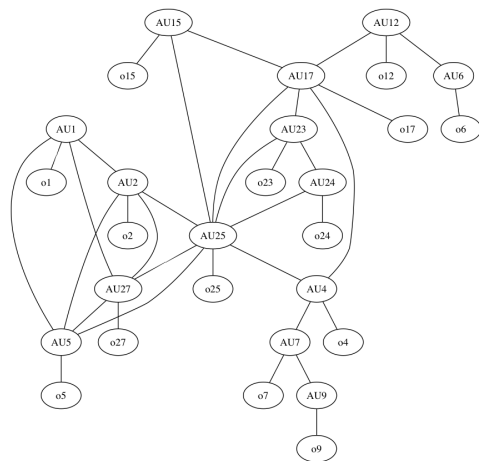

(a) MRF used in the computations.

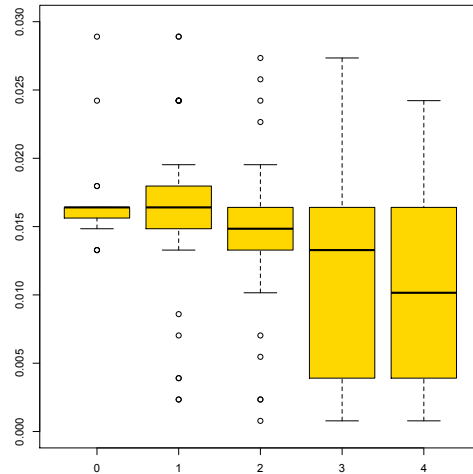

(b) Robustness split by Hamming distances.

Figure 1: On the left, the graph of the MRF used to compute MAP. On the right, boxplots for the robustness measure $\epsilon^*$ of MAP solutions, for different values of the Hamming distance to the truth.

vectors. Figure 1(b) shows the boxplot of $\epsilon^*$ for each value of the Hamming distance between 0 and 4 (lower $\epsilon^*$ of a MAP instantiation means lower robustness). As we can see in the figure, the median robustness $\epsilon^*$ decreases monotonically with the distance, indicating that this measure is correlated with the accuracy of the issued predictions, and hence can be used as a second order information about the obtained MAP instantiation for each instance.

The data set also contains information about the emotion expressed in the posed faces (at least for part of the images), which are shown in Figure 2(b): *anger, disgust, fear, happy, sadness and surprise*. We have partitioned the testing data according to these six emotions and plotted the robustness measure $\epsilon^*$ of them (Figure 2(a)). It is interesting to see the relation between robustness and emotions. Arguably, it is much easier to identify *surprise* (because of the stretched face and open mouth) than *anger* (because of the more restricted muscle movements defining it). Figure 2 corroborates with this statement, and suggests that the robustness measure $\epsilon^*$ can have further applications.

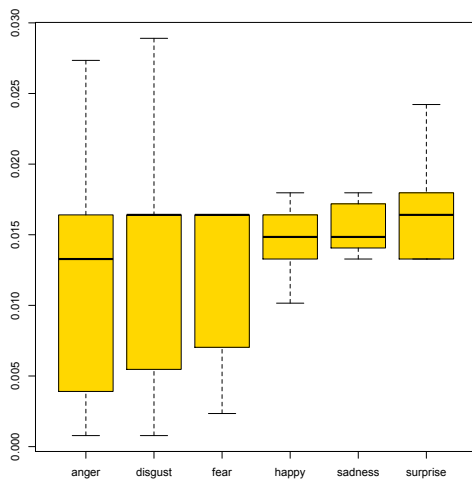

(a) Robustness split by emotions.

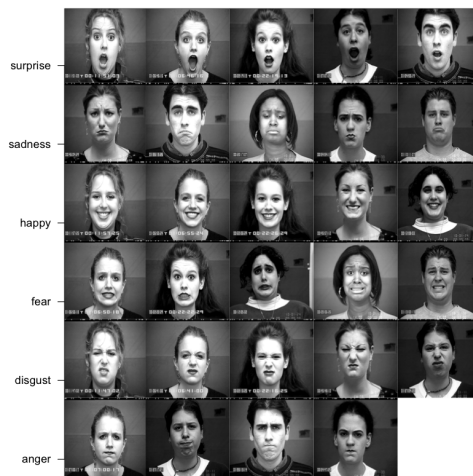

(b) Examples of emotions.

Figure 2: On the left, box plots for the robustness measure $\epsilon^*$ of the MAP solutions, split according to the emotion that was presented in the instance were MAP was computed. On the right, examples of emotions encoded in the data set [10, 16]. Each row is a different emotion.

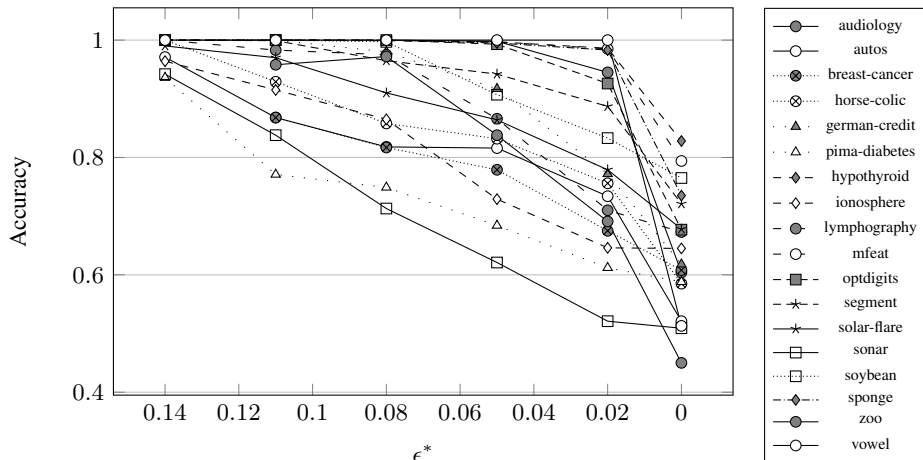

Figure 3: Average accuracy of a classifier over 10 times 5-fold cross-validation. Each instance is classified by a MAP inference. Instances are categorized by their $\epsilon^*$, which indicates their robustness (or amount of perturbation up to which the MAP instantiation remains unique).

## 5.2 Robustness of Classification

In this second experiment, we turn our attention to the classification problem using data sets from the UCI machine learning repository [1]. Data sets with many different characteristics have been used. Continuous variables have been discretized by their median before any other use of the data. Our empirical results are obtained out of 10 runs of 5-fold cross-validation (each run splits the data into folds randomly and in a stratified way), so the learning procedure of each classifier is called 50 times per data set. In all tests we have employed a Naive Bayes classifier with equivalent sample size equal to one. After the classifier is learned using 4 out of 5 folds, predictions for the other fold are issued based on the MAP solution, and the computation of the robustness measure $\epsilon^*$ is done. Here, the value $\epsilon^*$ is related to the size of the contamination of the model for which the classification result of a given test instance remains unique and unchanged (as described in Section 4.2). Figure 3 shows the classification accuracy for varying values of $\epsilon^*$ that were used to perturb the model (in order to obtain the curves, the technicality was to split the test instances into bins according to the computed value $\epsilon^*$, using intervals of length $10^{-2}$, that is, accuracy was calculated for every instance with $\epsilon^*$ between 0 and 0.01, then between 0.01 and 0.02, and so on). We can see a clear relation between accuracy and predicted robustness $\epsilon^*$. We remind that the computation of $\epsilon^*$ does not depend on the true MAP instantiation, which is only used to verify the accuracy. Again, the robustness measure provides a valuable information about the quality of the obtained MAP results.

## 6   Conclusions

We consider the sensitivity of the MAP instantiations of discrete PGMs with respect to perturbations of the parameters. Simultaneous perturbations of all the parameters (or any chosen subset of them) are allowed. An exact algorithm to check the robustness of the MAP instantiation with respect to the perturbations is derived. The worst-case time complexity is that of the original MAP inference times the number of variables in the domain. The algorithm is used to compute a robustness measure, related to changes in the MAP instantiation, which is applied to the prediction of facial action units and to classification problems. A strong association between that measure and accuracy is verified. As future work, we want to develop efficient algorithms to determine, if the result is not robust, what defines such instances and how this robustness can be used to improve classification accuracy.

### Acknowledgements

J. De Bock is a PhD Fellow of the Research Foundation Flanders (FWO) and he wishes to acknowledge its financial support. The work of C. P. de Campos has been mostly performed while he was with IDSIA and has been partially supported by the Swiss NSF grant 200021_146606 / 1.

## Footnotes

[1]Some authors refer to this problem as MPE (most probable explanation) rather than MAP.

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
