[Supplementary Material]

*Supplementary Material*

# Global Sensitivity Analysis
# for MAP Inference in Graphical Models

**Jasper De Bock**
Ghent University, SYSTeMS
Ghent (Belgium)
jasper.debock@ugent.be

**Cassio P. de Campos**
Queen's University
Belfast (UK)
c.decampos@qub.ac.uk

**Alessandro Antonucci**
IDSIA
Lugano (Switzerland)
alessandro@idsia.ch

## 1  Proof of Theorem 1

**Theorem 1.** *Let $X$ be a variable taking values in a finite set $\mathrm{Val}(X)$ and let $\mathcal{P}$ be a set of candidate mass functions over $X$. Let $\tilde{x}$ be a MAP instantiation for a mass funtion $P \in \mathcal{P}$. Then $\tilde{x}$ is the unique MAP instantiation for every $P' \in \mathcal{P}$ (equivalently $\mathrm{Val}^*(X)$ has cardinality one) if and only if*

$$\min_{P'\in\mathcal{P}} P'(\tilde{x}) > 0 \ \text{and} \ \max_{x\in\mathrm{Val}(X)\setminus\{\tilde{x}\}} \max_{P'\in\mathcal{P}} \frac{P'(x)}{P'(\tilde{x})} < 1, \tag{1}$$

*where the first inequality should be checked first because if it fails, then the left-hand side of the second inequality is ill-defined.*

*Proof.* We start by noticing that $\tilde{x}$ is the unique MAP instantiation for every $P' \in \mathcal{P}$ if and only if

$$\forall P' \in \mathcal{P} \,, \ \forall x \in \mathrm{Val}(X) \setminus \{\tilde{x}\} \ : \ P'(\tilde{x}) > P'(x). \tag{2}$$

In order for this condition to be satisfied, it is clearly necessary that $P'(\tilde{x})$ be strictly positive for each $P' \in \mathcal{P}$ or, equivalently, by the compactness of $\mathcal{P}$, that the leftmost part of Eq. (1) be satisfied. Under this condition, Eq. (2) can be rewritten as

$$\forall P' \in \mathcal{P} \,, \ \forall x \in \mathrm{Val}(X) \setminus \{\tilde{x}\} \ : \ \frac{P'(x)}{P'(\tilde{x})} < 1 \Leftrightarrow \max_{x\in\mathrm{Val}(X)\setminus\{\tilde{x}\}} \max_{P'\in\mathcal{P}} \frac{P'(x)}{P'(\tilde{x})} < 1, \tag{3}$$

where the compactness of $\mathcal{P}$ implies the existence of the final maximum. $\square$

## 2  Proof of Theorem 2

**Theorem 2.** *Let $\mathbf{X} = (X_1, \ldots, X_n)$ be a vector of variables taking values in their respective finite domains $\mathrm{Val}(X_1), \ldots, \mathrm{Val}(X_n)$, let $I_1, \ldots, I_m$ be a collection of index sets such that $I_1 \cup \cdots \cup I_m = [n]$ and, for every $k \in [m]$, let $\psi_k$ be a compact set of nonnegative factors over $\mathbf{X}_{I_k}$ such that $\Psi = \times_{k=1}^m \psi_k$ is a family of PGMs.*

*Consider now a PGM $\Phi \in \Psi$ and a MAP instantiation $\tilde{\mathbf{x}}$ for $P_\Phi$ and define, for every $k \in [m]$ and every $\mathbf{x}_{I_k} \in \mathrm{Val}(\mathbf{X}_{I_k})$:*

$$\alpha_k := \min_{\phi'_k\in\psi_k} \phi'_k(\tilde{\mathbf{x}}_k) \ \text{and} \ \beta_k(\mathbf{x}_{I_k}) := \max_{\phi'_k\in\psi_k} \frac{\phi'_k(\mathbf{x}_{I_k})}{\phi'_k(\tilde{\mathbf{x}}_{I_k})}. \tag{4}$$

*Then $\tilde{\mathbf{x}}$ is the unique MAP instantiation for every $P' \in \mathcal{P}_\Psi$ if and only if*

$$(\forall k \in [m]) \ \alpha_k > 0 \ \text{and} \ \prod_{k=1}^m \beta_k(\mathbf{x}_{I_k}^{(2)}) < 1, \tag{RMAP}$$

*where* $\mathbf{x}^{(2)}$ *is an arbitrary second best MAP instantiation for the distribution* $P_{\tilde{\Phi}}$ *that corresponds to the PGM* $\tilde{\Phi} := \{\beta_1, \ldots, \beta_m\}$*. The first criterion in* (RMAP) *should be checked first because* $\beta_k(\mathbf{x}_{I_k}^{(2)})$ *is ill-defined if* $\alpha_k = 0$.

*Proof.* Since every set of factors $\psi_k$ is compact, $\mathcal{P}_{\Psi}$ is compact as well. Therefore, by Th. 1, $\tilde{\mathbf{x}}$ is the unique MAP instantiation for every $P' \in \mathcal{P}_{\Psi}$ if and only if

$$\min_{P' \in \mathcal{P}_{\Psi}} P'(\tilde{\mathbf{x}}) > 0 \text{ and } \max_{\mathbf{x} \in \mathrm{Val}(\mathbf{X}) \backslash \{\tilde{\mathbf{x}}\}} \max_{P' \in \mathcal{P}_{\Psi}} \frac{P'(\mathbf{x})}{P'(\tilde{\mathbf{x}})} < 1. \tag{5}$$

Hence, we are left to prove that Eq. (5) is equivalent to (RMAP). By the compactness of $\mathcal{P}_{\Psi}$:

$$\min_{P' \in \mathcal{P}_{\Psi}} P'(\tilde{\mathbf{x}}) > 0 \Leftrightarrow (\forall P' \in \mathcal{P}_{\Psi}) \, P'(\tilde{\mathbf{x}}) > 0 \Leftrightarrow (\forall \Phi' \in \Psi) \, P_{\Phi'}(\tilde{\mathbf{x}}) > 0$$

$$\Leftrightarrow (\forall \Phi' \in \Psi) \, \frac{1}{Z_{\Phi'}} \prod_{k=1}^{m} \phi_k'(\tilde{\mathbf{x}}_{I_k}) > 0 \Leftrightarrow (\forall \Phi' \in \Psi) \, \prod_{k=1}^{m} \phi_k'(\tilde{\mathbf{x}}_{I_k}) > 0$$

$$\Leftrightarrow (\forall \Phi' \in \Psi)(\forall k \in [m]) \, \phi_k'(\tilde{\mathbf{x}}_{I_k}) > 0 \Leftrightarrow (\forall k \in [m])(\forall \phi_k' \in \psi_k) \, \phi_k'(\tilde{\mathbf{x}}_{I_k}) > 0.$$

Thus, given the compactness of the sets $\psi_k$, the first inequality in Eq. (5) is equivalent to the first criterion in (RMAP).

If this first criterion holds, again using the compactness of the sets $\psi_k$, we find that all the $\beta_k(\mathbf{x}_{I_k})$ are well-defined and nonnegative. Also, if the first criterion holds, then for all $\mathbf{x} \in \mathrm{Val}(\mathbf{X})$:

$$f(\mathbf{x}) := \max_{P' \in \mathcal{P}_{\Psi}} \frac{P'(\mathbf{x})}{P'(\tilde{\mathbf{x}})} = \max_{\Phi' \in \Psi} \frac{P_{\Phi'}(\mathbf{x})}{P_{\Phi'}(\tilde{\mathbf{x}})} = \max_{\Phi' \in \Psi} \prod_{k=1}^{m} \frac{\phi_k'(\mathbf{x}_{I_k})}{\phi_k'(\tilde{\mathbf{x}}_{I_k})} = \prod_{k=1}^{m} \max_{\phi_k' \in \psi_k} \frac{\phi_k'(\mathbf{x}_{I_k})}{\phi_k'(\tilde{\mathbf{x}}_{I_k})} = \prod_{k=1}^{m} \beta_k(\mathbf{x}_{I_k}).$$

Thus, since $f(\tilde{\mathbf{x}}) = 1$, $\tilde{\Phi} = \{\beta_1, \ldots, \beta_m\}$ is indeed a PGM. To conclude the proof, we show that the second inequality in Eq. (5), which can now be reformulated as

$$c := \max_{\mathbf{x} \in \mathrm{Val}(\mathbf{X}) \backslash \{\tilde{\mathbf{x}}\}} f(\mathbf{x}) < 1,$$

is equivalent to $f(\mathbf{x}^{(2)}) < 1$. Let $\mathbf{x}^{(1)}$ be (one of) the MAP instantiation(s) for $P_{\tilde{\Phi}}$ that enable(s) $\mathbf{x}^{(2)}$ to satisfy Eq. (5,main paper). First, assume that $f(\mathbf{x}^{(2)}) < 1$. Then by Eq. (5,main paper) and because $f(\tilde{\mathbf{x}}) = 1$, we see that $\mathbf{x}^{(1)} = \tilde{\mathbf{x}}$ and therefore that $c = f(\mathbf{x}^{(2)}) < 1$. Next, assume that $c < 1$. Then by Eq. (5,main paper) and because $f(\tilde{\mathbf{x}}) = 1$, we find that $\mathbf{x}^{(1)} = \tilde{\mathbf{x}}$ and $f(\mathbf{x}^{(2)}) < 1$. $\qquad \square$