[Reviews · NeurIPS 2014]

Submitted by Assigned_Reviewer_15

The paper gives a nice introduction to sensitivity analysis in graphical models, and proposes a method to check if the MAP configuration will not change, if a simultaneous perturbation in the parameters occurs. The method works by selecting perturbed factors in such a way (using local computations) to create a worst-case scenario. The second best MAP estimation is then identified for the worst case scenario. Theoretically, the complexity is therefore equivalent to the MAP query (exponential in the tree-width), if the number of variables is a constant. As presented, one disadvantage of this technique is that, practically, it requires computational time: n*[MAP inference time], where n is the number of variables. For non-trivial graphical models, the proposed sensitivity analysis can be orders of magnitude slower than the already hard MAP query. That said, more efficient algorithms for getting the M-Best MAP seem to exist and the authors may consider to utilize one of them as a black box. See for example: An Efficient Message-Passing Algorithm for the M-Best MAP Problem, by Dhruv Batra. See also: Search Algorithms for m Best Solutions for Graphical Models, by Rina Dechter et al. On the other hand, the experimental results in the paper do not provide very useful information, as discussed below.

Quality:

The paper appears to be technically correct. Theorem 1 is trivial and could be a lemma for Theorem 2. The proof of Theorem 2 in the supplementary material could be re-written in a clearer way to provide more insight. For example, you could discuss the choice of factors in the new PGM as creating a worst-case, and briefly mention that the partition function cancels out and has no effect.

On the other hand, the experimental results show that the robustness measure is a good indication for the testing accuracy. While I see why there is a relation between robustness and testing accuracy, I think the relation is much more involved and may be linked to the bias-variance trade-off. Thus, the experiments seem to be distantly related to the current work and not the best choice, or perhaps not well-motivated. I have to admit though that it might be hard to design very meaningful experimental results in this case. The authors could perhaps verify the proof experimentally and show how pessimistic the analysis is, in real-world applications, by having an experimental measure of sensitivity.

Clarity:

The paper is clearly written when it talks about preliminaries, and related work. When it comes to the contribution, it lacks insightful explanation.

Originality:

To the best of my knowledge, the contribution is new.

Significance:

The paper tackles an important problem. Most of the technical results in the paper do not appear to be so surprising though. Worst-case analysis is widely used in many fields. If a system passes the worse case, it passes all cases. While the paper proposed a novel exact method to do such analysis, in the context of MAP estimation, the inefficiency of the method, and the lack of meaningful experimental results have negatively affected the quality of the work.

On the other hand, in some cases, the proposed analysis may be too pessimistic. It might be interesting to try to give a probabilistic guarantee that is more efficient to compute.

Other comments/suggestions:

In Figure 1 (a): The font is too small.
In Figure 1 (b): I suggest adding labels for the axes, and on the plot (maximum, minimum, ...etc).
Equation 5: Did you forget to mention that x(1) is the best MAP?

UPDATE:
*******
After reading the authors' response, I am convinced that part of my criticism may have been a bit philosophical, and, therefore, should not lead to rejecting the paper. I am glad to raise my score, and hope the authors can get their final version in the best form, if the paper is accepted.

Summary: The paper appears to be technically correct. The inefficiency of the proposed sensitivity analysis has affected the evaluation negatively. The experiments do not provide very useful information.

Submitted by Assigned_Reviewer_27

The authors consider a sensitivity analysis of the MAP inference problem for graphical models. In particular, they consider the question of by what factor parameters (weights/conditional probabilities) of a graphical model can be perturbed while still maintaining the same MAP solution. This is a interesting and important problem. For example, the sensitivity analysis (SA), it can help us decide whether or not more training data is needed to tune parameters in graphical models, and also which parameters may need further tuning.

The authors then show how SA can be stated as an optimization problem (hmm... now I realize why SA is a bit a tricky abbreviation in this context... one could us SA, as in simulated annealing, to solve the optimization problem...) with essentially the same complexity of the MAP task itself. This is actually somewhat surprising because the problem "feels" more difficult (you have to decide for all possible perturbations of parameters within a specified range whether the MAP solution stays the same). It would be good to state the formal complexity of the optimization problem vs. the MAP problem itself. (How much is hidden in"essentially" qualifier?)

The paper is very well-written. The examples are hugely beneficial. I did not check the full technical details but the results appear valid. The experimental results support the formal framework and are interesting.
Summary: The authors provide a new approach for parameter sensitivity analysis (SA) for graphical models wrt MAP inference. Somewhat surprisingly, they show that SA can be formulated as an optimization task with essentially the same complexity as MAP itself. Novel and interesting results.

Submitted by Assigned_Reviewer_41

This paper is interested in the sensitivity of MAP explanations in
probabilistic graphical models, subject to perturbations in the
network parameters. That is, how robust is the most likely assignment
to changes in the network parameters. The authors perspective is
based on viewing the problem as one of finding the most likely
instantiations in a family of graphical models (representing the
perturbations). In particular, the authors construct a new model,
where finding the second-most likely instantiation determines whether
there exists a unique maximizing instantiation in the family, or not.
In this way, one can grow this family (induce greater perturbations)
until a new maximizing instantiation is found, indicating the
robustness of the MAP explanation in the original model.

I view sensitivity analysis as a fundamental problem in probabilistic
graphical models, although in many cases it is computationally
intractable. Multi-parameter sensitivity analysis, for example, is
particularly hard (see Kwisthout & van der Gaag UAI'08). The current
paper provides a novel perspective on the problem, in the context of
MAP explanations.

As far as qualitative approaches to sensitivity analysis, there is at least
one more notable example:

"Evidence and scenario sensitivities in naive Bayesian classifiers"
by Silja Renooij, Linda C. van der Gaag, in IJAR 2008

which considers the robustness of a classification with respect to
changes in a single network parameter, in contrast to global changes
as in the current paper. The authors also consider naive Bayes
networks in the experiments (at least, I believe the settings are
equivalent here, if we are maximizing a single unobserved variable
here, i.e., the class). It may be interesting to see the differences
here, or whether different conclusions could be drawn about the
robustness of naive Bayes classifiers, when one considers global
parameter changes; or at least, one could view the current paper as
generalizing the results of the paper above (assuming again, the
setting is indeed the same).

Another related paper is:

"Algorithms and Applications for the Same-Decision Probability"
by Suming Jeremiah Chen, Arthur Choi, Adnan Darwiche, in JAIR 2014

which can be viewed (roughly) as analyzing the sensitivity of a
decision, against features that have yet to be observed.

There has also been recent interest in approaches to probabilistic
inference based on MAP perturbations, e.g., (Hazan & Jaakkola ICML'12)
and (Papandreou & Yuille ICCV'11). It may be interesting to if any of
the techniques here could be applied to analyzing these approaches.
Summary: The paper provides an interesting view on sensitivity analysis for MAP
explanations in probabilistic graphical models.
Author Feedback
Author rebuttal: We thank all reviewers for their thoughtful comments and suggestions. Responses to their main comments are provided below.

Assigned_Reviewer_15:

We appreciate the references to literature on finding the second best MAP. We are aware of them, but we consider (for the purpose of this paper) that "n*[MAP inference time]" is efficient enough. We recall that MAP inference can be effectively run in networks with hundreds (or even thousands) of variables very quickly (there are hard cases too, obviously), and that the effort of the community to improve MAP solvers has been very successful in the recent years. Any speed improvement for MAP inferences will be an automatic improvement for our global sensitivity analysis too. Also, while we work with exact methods for MAP and its sensitivity analysis, approximate methods for MAP and second-best MAP could be used as well. Our main idea of reducing the problem of global sensitivity analysis in PGMs to the computation of second best MAP holds still.

There are a number of differences between standard worst-case analysis techniques and our results. For instance, worst-case analysis is often quantitative in nature, and often is interested in the worst value of some numerical parameter (for example, the failure probability), thereby being an optimization problem by nature. In our case, the problem is more of a "qualitative" nature (does the MAP solution change?). One of our contributions is to show that it is nevertheless possible to transform this problem into an optimization task. We believe that this result is valuable in itself, and it is for this reason that we have turned it into a separate theorem (despite the fact that, indeed, the proof of this Theorem 1 is rather trivial). Theorem 2 then goes on to solve this optimization problem for the particular case of PGMs.

There is obviously a relation of classification accuracy and the bias-variance trade-off. However, we do not study such phenomenon here. We take on a different avenue, which is somewhat expected too: accuracy of a MAP inference is related to how much the MAP solution is "robust" to variations of the underlying model. Our approach provides a robustness measure for each issued MAP solution, so one can use this measure as a second-order information about the MAP solution.

The relation between global robustness of a MAP inference (in the sense that we present in this work, that is, how much the model would need to be globally perturbed until another MAP solution would emerge) and accuracy of classification (or accuracy of the MAP solution) could not be verified before, as no method in the literature could perform such analysis in a reasonable time, that is, no previous work had been able to study the global sensitivity of MAP in probabilistic graphical models in a way that the user can freely choose which parameters of the PGM they want to vary (one parameter, some parameters, all parameters, different variations, etc). We believe that our (choices of) experiments are justified by the fact that they clearly illustrate this relation.

Assigned_Reviewer_27:

Our current implementation is n times the cost of a MAP inference, where n is the number of variables. There are no large hidden constants/exponents, apart from the time to compute the local parameters that are given to the second-best inference. Since, usually, we are interested in linear/convex constraints only and such constraints are local, this step is very fast. This means that one can run this global sensitivity analysis approach whenever MAP inference can be run in a reasonable amount of time. Furthermore, as the reviewers point out, other techniques for finding the second best MAP could be used to improve the performance even more.

Assigned_Reviewer_41:

We will put the cited papers into perspective with our work. While part of our experiments use the Naive Bayes classifier, the techniques we present are not restricted to it. So it can be used as a global sensitivity version of Renooij et al. 2008 (and we will properly refer to that work), but it also goes further to querying any number of MAP variables in any network topology. On the other hand, Chen et al. 2014 take a different approach to robustness, based on the sensitivity w.r.t. data/new observations (as the reviewer also points out). We will consider mentioning it (as well as the other references), but that would be appropriate only if we can link it with sensitivity as we intend it in this paper.